# Long-Latency Event-Related Potentials (300–1000 ms) of the Visual Insight [note 1]

**DOI:** 10.3390/s22041323

**Published:** 2022-02-09

**Authors:** Sergey Lytaev

**Affiliations:** 1Department of Normal Physiology, St. Petersburg State Pediatric Medical University, 194100 Saint Petersburg, Russia; mail@physiolog.spb.ru; Tel.: +7-921-938-5120; 2Lab of Applied Informatics, St. Petersburg Federal Research Center of the Russian Academy of Sciences, 199178 Saint Petersburg, Russia

**Keywords:** cognition, event-related potentials, wave P_300_, late components of ERPs, oddball paradigm, sensory recognition, images-illusions

## Abstract

The line of insight research methods that have high temporal and surface resolution is not large—these are EEGs, EPs, and fMRI, as well as their combinations and various options for assessing temporal events of random understanding. The objective of this research was to study the classification of insight for visual illusory images consisting of several objects simultaneously according to the analysis of early, middle, late, and ultra-late components (up to 1000 ms) of event-related potentials (ERPs). ERP research on 42 healthy subjects (men) aged 20–28 years was performed. The stimuli were a line of visual images with an incomplete set of signs, as well as images-illusions, which, with different perceptions, represent different images. The results showed the similarity of the tests to correct recognition of fragments of unrecognition and double images. At the intermediate stage of perception (100–200 ms), in both cases, the activity of the central and frontal cortex decreased, mainly in the left hemisphere. At the later stages of information processing (300–500 ms), the temporal-parietal and occipital brain parts on the right were activated, with the difference that when double objects were perceived, this process expanded to 700–800 ms with the activation of the central and occipital fields of the right hemisphere. Outcomes allowed discussing two possible options for actualizing the mechanisms of long-term memory that ensure the formation of insight—the simultaneous perception of images as part of an illusion. The first of them is associated with the inhibition of the frontal cortex at the stage of synthesis of information flows, with the subsequent activation of the occipital brain parts. The second variant is traditional and manifests itself in the activation of the frontal brain areas, with the subsequent excitation of all brain fields by the mechanisms of exhaustive search.

## 1. Introduction

There is a line of definitions of insight, depending on the selected combination of characteristics or, rather, on the objectives of the study—psychological, clinical, physiological, marketing, etc. [1,2,3,4,5]. Typically, the definition is associated with a sudden solution to a problem, which is preceded by an impasse and restructuring of the problem, followed by a positive emotional reaction. There is a mismatch in cognitive psychology and neuroscience, which we consider to be the main criterion for understanding, namely surprise. For example, in a series of studies on insight, the authors do not specifically highlight decisions that have arisen suddenly [6,7,8]. Another widespread use of the term “insight” can be found in clinical psychology, where understanding refers to self-awareness, often personal symptoms, functional impairment, or other types of foresight. The clinical and unscientific use of the term does not require sudden awareness or any concomitant emotional response. Indeed, in clinical psychology, the absence of an emotional response can itself be seen as a symptom indicating a lack of understanding [9,10,11].

Currently, there are no universal scales that measure the variability of insight perception. Scales have been created to measure factors that are believed to be key drivers of insight. In particular, it is proposed that the radicality of insight (the perceived deviation between previous and new ideas about problems) and the restructuring of experience (the subjective experience of restructuring the representation of a problem) be considered [12].

The mechanisms for the benefits of memory in the process of insight, which are best supported by the results, are that these mechanisms are a joint consequence of finding the right solutions, the subjective feeling that the person has found the right solution (confidence), and the experience of an emotional pleasant reaction (reward) while solving the problem. This is the process that all contributes to better memory for the solution [13].

In particular, there is a definition of insight as any sudden understanding, awareness, or solution of a problem, which includes reorganizing the elements of a person’s mental representation of a stimulus, situation, or event to obtain a non-obvious or non-dominant interpretation [11]. More recently, the responsibility for gross semantic coding and internal focused attention before and during the solution to the problem of insight has been established in the right hemisphere. Individual differences in the propensity to solve problems in an insightful rather than conscious and analytical way are associated with different patterns of brain activity at rest. Direct brain stimulation has also recently come to be used to facilitate understanding.

It is known that a number of conscious and unconscious effects are used to improve the marketing efficiency of an advertising campaign. However, little is known about the individual contribution that conscious and unconscious processes make to the cognitive effectiveness of creative advertising. Some of these processes can also trigger insights.

The analysis of such studies shows that the moment of insight associated with advertising, whether it is standard or creative, consolidates unconscious memory, while non-insightful advertising improves conscious memory. It is widely accepted that insight decisions carry the Aha! experience. Such experiences are better remembered than those that lack a sense of insight. One question in recent research is whether the memory advantage for problem solving is modulated by the affective component of insight (the strong emotions usually accompanying the Aha! experience) or the cognitive component of insight in the form of a restructuring or change in representation that occurs during insightful problem solving [14,15].

Evaluation of the effectiveness of direct and indirect advertising is one of the tasks of applied research in assessing the mechanisms of insight. Direct advertising openly portrays the advertised products and brands. In indirect advertising, the advertising message needs to be improved. One of the methods for assessing the difference in the perception of advertising products is registration of eye movements. Overall, it was found that indirect advertising images received more fixation, as did indirect brand preference. Indirect brand logo recognition improved when tested after a shorter delay. Most interestingly, consumers perceived indirect advertising as more original, surprising, intelligent, and more difficult to interpret than direct advertising. The results showed that indirect advertising induces cognitive rework, leading to higher preference and memorability for brands [16].

It is believed that sudden insight while solving a problem can improve learning, but the underlying neural processes are largely undeveloped. For this purpose, the neural correlates of learning based on the sudden understanding of a verbal task based on the data of functional magnetic resonance imaging have been investigated. The findings suggested that learning by sudden comprehension may constitute one of the special cases where new information is directly encoded into semantic memory, similar to previous records of schema-dependent memory or prior knowledge. In addition, the results also provide further insight into how mnemonic schemas can be quickly generated from previously remotely related semantic representations [17,18].

Unconscious information can influence our behavior both in an experimental context and in life conditions. The reverse question is whether these effects can be applied in the context of everyday life (e.g., in advertising), which is a subject of serious controversy. Research shows that if real information is minimally presented at a conscious or even unconscious level, it can affect our subsequent behavior, even if more than five seconds elapse between the presentation of minimally conscious or unconscious information and the behavior that it affects [19].

Thus, the cognitive neurobiology of insight is an exciting new area of research related to fundamental neurocognitive processes [20,21,22]. The article discusses the results of original research in the context of the discussion of modern studies of the well-known psychological phenomenon of P_300_ visual evoked potentials (VEPs). The objective of this research was to study the classification of insight for visual illusory images consisting of several objects simultaneously according to the analysis of early, middle, late, and ultra-late components (up to 1000 ms) of event-related potentials.

## 2. Material and Methods

Research on 42 healthy subjects (men) aged 20–28 years was performed. Event-related potentials (ERPs) traditionally in 19 monopolar sites were recorded with reference electrodes on the earlobes. Accordingly, the dipole was the signal between the scalp and reference electrodes on each side. The signal sampling rate was –500 Hz, and the amplifier bandwidth was 0.5–30 Hz. These are the traditional parameters for recording long-latency evoked potentials (EPs) and EEGs. The purity of the signal was determined by automatic indicators. The registration of biosignals began with the registration of a traditional EEG. After that, the amplifier was switched to registration of super-averaged VEPs.

Visual images for a short time (duration 0.3 ms, frequency 0.5 Hz, 30 savings) on the display screen were presented. The stimuli were a line of visual images with an incomplete set of signs, as well as images-illusions, which, with different perceptions, represent different images. Dual visual images-illusions with complete correct identification were a model of the phenomenon of insight [23].

A dual image “vase face candlestick” was presented on the monitor screen as a test object. The subjects received the instruction: “The image of the 1st or several figures will be rhythmically shown on the screen in front of you. You need to memorize them and, after the end of the study, submit a report on the figures that you memorized.” The use of dual images-illusions as a model made it possible, on the one hand, to fix all three possible answers (1: not a single figure was identified; 2: one figure was identified; and 3: everything) and, on the other hand, on this basis, to fix relatively homogeneous groups for mathematical analysis. A stepwise discriminant analysis was carried out in comparison with the data of a control group of healthy subjects, where ERP registration was performed in response to a traditionally recognized visual image (key fragments) [8].

The recognition of one of the dual visual images, as well as the recognition of images with an incomplete set of features, presented a model of perception of oddball images. The lack of correct identification was considered as a common figurative stimulation. The quantitative characteristics of the answer options in the recognition of dual visual images-illusions and fragments of oddball images are presented in Table 1.

The P_300_ wave, as well as the later waves N_450_, N_750_, and N_900_, ERPs in solving ergonomic problems can be considered indicators of the categorization of the perception of visual illusions. The baseline amplitude and peak latency (PL) automatically in 2 ms steps were measured. Further, the amplitude-time characteristics were transferred to the big data processing system, where stepwise discriminant and factor analyses to establish the stability of ERP parameters were applied. Differences were considered statistically significant when the value of the F-statistic of the stepwise discriminant analysis was more than 4 (F > 4.0) or the discriminant function (DF) was calculated. A stepwise discriminant analysis was carried out in comparison with the data of a control group of healthy subjects with VEP registration in response to an identifiable visual image.

## 3. Results

Visual event-related potential in response to visual imagery stimulation at an analysis epoch of 1000 ms had the following typical pattern (see Figure 1). Mainly, the maximum activation in the brain posterior parts was recorded. In addition, we analyzed early negative waves with a peak latency (PL) of 50–100 ms (N_70_), intermediate components having a maximum amplitude at PL values of 120–200 ms (N_150_), and late negative fluctuations in the interval of the analyzed epoch of 300–450 ms (N_350_ and N_450_). After that, in the dynamics of VEPs, there was a certain lull corresponding to a time interval of 450–700 ms and representing an averaged electroencephalogram. Following this “quiet” area, VEP waves also were noted, sometimes comparable in amplitude with the intermediate components, and named by us N_750_ and N_900_. Of the positive waves, only an oscillation with a PL of 200–300 ms (P_250_) was analyzed, which, among other components located below the isoline, had the maximum amplitude and a more stable manifestation.

The N_70_ and N_150_ components changed their parameters relatively insignificantly in this study (see Table 2). At the same time, it can be noted that the N_70_ amplitude in the occipital leads (F > 4.0) reduced during the perception of glass fragments, and the N_150_ amplitude decreased in the left hemisphere in the central (F = 8.0) and both frontal registration points. In the case of correct recognition of dual images, the N_70_ parameters did not differ from the control ones (F < 4.0), but by analogy with the perception of glasses, the N_150_ amplitude reduced in leads C_3_ (F = 6.4, DF) and both frontal leads, with a predominance in the left half of the brain (F = 6.1). Correct detection of one figure was accompanied by a lightening of the N_70_ amplitude in the right frontal sites (F = 9.9), and in the absence of recognition, the N_70_ amplitude increased in the entire frontal cortex. In this case, the characteristics of the N_150_ did not change (F < 4.0).

The error-free detection of glasses was characterized by an increase in the P_250_ amplitude in all areas of the neocortex, but it was significantly recorded only in the right frontal area (F = 5.4, DF). The parameters of the N_350_ wave in this group of subjects did not differ from the control ones (F < 4.0). Correct detection of dual images did not change the P_250_ amplitude but contributed to a noticeable shortening of P_250_ PL throughout the neocortex (F > 4.0). The N_350_ amplitude increased in the posterior parts of the cortex (F > 4.0), and a shortening of the N_350_ PL was also noted here. With the correct detection of one of the dual objects, the peak latency of P_250_ and N_350_ was shortened at almost all registration points (F > 4.0). The amplitude of P_250_ increased in the central (F = 4.1) and frontal (F = 5.1) leads of the right hemisphere, and N_350_ in the left occiput (F = 4.2). In the absence of recognition, the PL and the P250 amplitude did not differ from the control ones (F < 4.0), and the N_350_ amplitude reduced mainly in the anterior half of the brain (F > 4.0); see Table 3.

Later waves changed insignificantly with this type of identification. Only a slight increase in amplitude was noted: N_450_ in the left occiput (F = 4.4) and N_750_ in the occipital (F = 5.2) and central (F = 4.4) leads of the right hemisphere. The N_900_ parameters remained unchanged from the control parameters (F < 4.0); see Table 4, Table 5 and Table 6.

Correct detection of glass fragments did not affect the parameters of the latest (N_900_) wave (see Table 6). In this case, the N_450_ amplitude increased in the occipital cortex (F > 4.0), while its PL increased in the frontal areas (F > 4.0); see Table 4. The N_750_ amplitude did not differ from the control one (F < 4.0), but the PL reduced throughout the entire cortex (F > 4.0); see Table 5. Recognition of one of the dual images did not affect the characteristics of N_450_, but in almost all leads, the amplitude of N_750_ increased (F > 4.0) with a maximum in the right frontal cortex (F = 17.3, DF). The amplitude of N_900_ increased only in the right occiput (F = 4.8) and right frontal points (F = 7.5), as well as in leads C3 (F = 9.2, DF) and F3 (F = 6.3). In cases of lack of recognition upon presentation of dual images, the amplitude of N_450_ increased throughout the neocortex (F > 4.0), while the amplitude of N_750_ and N_900_ increased in all leads (F > 4.0), except for the occipital (F < 4.0).

Later waves with full recognition of images-illusions changed as follows (see Table 4). The amplitude of N_450_ in the left occiput increased (F = 4.4), and the amplitude of N_750_ also increased in the occipital (F = 5.2) and central (F = 4.4) sites of the right hemisphere. The N_900_ parameters remained unchanged from the control data (F < 4.0); see Table 6.

Correct detection of oddball patterns did does not affect the parameters of the latest (N_900_) wave (see Table 6). At the same time, the N_450_ amplitude increased in the occipital cortex (F > 4.0), while the PL increased in the anterior regions (F > 4.0); see Table 4. The N_750_ amplitude did not differ from the control one (F < 4.0), but the PL reduced throughout the cortex (F > 4.0); see Table 5.

Recognition of one of the dual images did not affect the characteristics of N_450_, but in almost all sites, the amplitude of N_750_ increased (F > 4.0), with a maximum in the right frontal cortex (F = 17.3). The amplitude of N_900_ increased only in the right occiput (F = 4.8) and frontal (F = 7.5) points, as well as in sites C_3_ (F = 9.2) and F_3_ (F = 6.3); see Table 4–6.

Unrecognition upon presentation of dual visual images-illusions, the amplitude of N_450_ increased in a generalized manner (F > 4.0; see Table 4) and the amplitude of N_750_ and N_900_ increased in all leads (F > 4.0), except for the occipital registration points (F < 4.0); see Table 5 and Table 6.

Figure 2, Figure 3 and Figure 4 summarize the main statistical findings. A particular typical example of the spatial distribution of VEP is presented with the correct recognition of dual visual images-illusions (Figure 2). There was a reduction in evoked activity (especially at the early and intermediate stages of information processing) in the frontal and central regions of the brain, as well as an increase in the amplitude of the N_350_ wave in the temporal-parietal regions of the right hemisphere (O_2_, P_4_, T_6_). Figure 3 reflects a typical example of the VEP spatial distribution in the case of identifying one of the dual figures—the oddball image variant. There was an activation of the frontal cortex (F_3_, Fz, F_4_), as well as a well-pronounced component N_450_, by analogy with the variants of the lack of recognition and activation of short-term visual memory. The lack of recognition of visual images was characterized by generalized symmetric activation throughout the entire epoch of analysis, more pronounced in the posterior half of the brain (Figure 4).

## 4. Discussion

Despite the existing psychological reasons for the hypothesis that the effect of insight is theoretically and functionally important, much less research has been carried out on its neural basis than psychological experiments [17]. Insight as a psychological phenomenon has been studied for at least 100 years, but the methods of neuroimaging and neurophysiology in the study of insight began to be used only in the past 20–30 years [20,21,23,24,25]. In summary, the set of research methods is not large—these are EEGs, EPs, and fMRI, as well as their combinations and various options for assessing temporal events of random understanding, the moment of Aha! [26].

Neuroimaging studies of insight are carried out mainly based on EEG data. Other methods of brain research are less common [21,27,28]. The EEG is considered to have high temporal resolution with limited spatial resolution. fMRI, on the other hand, has excellent spatial resolution with limited temporal resolution, so it is better suited for localizing a neural event in space. Together, these methods are able to isolate neural correlates of insight in both space and time. The combination of techniques is critical because fMRI’s ability to localize insight-related neural activity would be less informative if it were not for knowing whether these neural correlates arose before, after, or at the time of decision [9].

When the problem is solved intuitively, the EEG shows a burst of high-frequency EEG gamma activity over the right temporal lobe and fMRI shows a corresponding change in blood flow in the right anterior superior temporal gyrus [9]. In the original fMRI experiment, this right temporal region was the only area that exceeded the statistical thresholds. Further, weak activity was found in the hippocampus and parahippocampal gyri, as well as in the anterior and posterior cingulate cortex. In later research, the same network of regions far exceeded the critical statistical threshold, with the right anterior temporal region again being the strongest [5].

The results of this study of ERP registration in response to ambiguous images of illusion show the similarity of the tests to correct recognition of fragments of glasses and double images. At the intermediate stage of perception (100–200 ms), in both cases, the activity of the central and frontal cortex decreases, mainly in the left hemisphere. At the later stages of information processing (300–500 ms), the temporo-parietal and occipital parts of the brain on the right are activated, with the difference that when double objects are perceived, this process expands to 700–800 ms with the activation of the central and occipital fields of the right hemisphere.

It is known that the area of interaction of the inferior temporal and posterior parietal cortex is the secondary projection zone of the visual system. It is here that the formation (but not recognition) of visual images, complex scenes, frames, frames, etc., takes place. Final recognition is a more complex process involving the associative frontal cortex of the cerebral hemispheres, as well as the reward system. It should be noted that an important mechanism for coordinating the functioning of these departments is the inhibition reaction [29]. This response is the essence of various neurological disorders, such as schizophrenia and attention deficit hyperactivity disorder, which are also targeted by insight research.

Other electrophysiological data suggest that even from two opposite ERP results, it is possible to trace the features of internal and external insight [13]. Internal insight is associated with positive ERP components after stimulus onset (P200-600) above the superior temporal gyrus [15]. External insight is associated with a negative component of ERP (N320) [16]. These results already show that external and internal insights differ at the behavioral and neurophysiological levels. Various neurobiological insights suggest that presenting a solution or a decision clue leads to the same Aha! moment as attempting an inner decision [30,31,32,33,34,35].

The spatial and temporal consistency of fMRI and EEG results suggested that these findings are caused by the same underlying brain activation [9,21]. The response of the right temporal brain was identified as the main neural correlate of the experience of insight, because (a) it occurred approximately at the moment when the participants realized the solution to each of these problems, (b) the same area is involved in other tasks requiring semantic analysis and integration, and (c) gamma-band activity has been proposed as a mechanism for binding conscious information.

This assumption is supported by other neurophysiological data [9]. Significantly more activity in the alpha range was found before analysis than before an analytical report was obtained on the parietal electrodes on the right. The authors localized this effect of internal insight in the time interval from −1310 ms up to −560 ms relative to the solution. In both intrinsic and extrinsic insight effects, no indicators of some error on time were found due to different response times. Moreover, repeated-measures ANOVA confirmed the hypothesis that internal and external insights have the opposite effect on alpha power, with significant cross talk between the problem-solving strategy, that is, insight or analysis, and range, generating or recognizing a solution.

In the problem of sudden insight, the neural correlates of learning using fMRI and the verbal problem of problem solving were investigated. Activation of the hippocampus, medial prefrontal cortex (mPFC), amygdala, and striatum was found during sudden awareness. Remarkably, mPFC and temporo-parietal structures, rather than the hippocampus, were associated with later learning to sudden decision making. At the same time, complex traditional insight, compared to light flash insights, triggered stem activation at the midbrain level and was associated with successful learning through intrinsic reward [17,18].

Participants’ neural responses to induced, possibly insight-like, sudden understandings where solutions to complex problems for remote workers were revealed to them after a short delay were measured. Increased activity has been found in the striatum, part of the brain’s reward system, as well as in the amygdala, hippocampus, and medial prefrontal cortex, suggesting a link between the hypothetical neural reward and its integration into memory [17].

While the participants were solving complex problems of remote partners, we used an ultrahigh magnetic field (7T) to obtain high-resolution fMRI images in the subcortical regions [36]. Evidence has been found for insight-related brain activity in the dopaminergic regions of the nucleus accumbens and the ventral region (VTA), implying that insight engages neural reward mechanisms. However, the specifics of the procedure of this study limited the scope of its results. For example, instead of asking subjects to report whether each solution was obtained through insight or analysis, they were asked to “estimate the amount of insight and deadlock they experienced while completing the task.” This instruction confused understanding and deadlock and did not separate understanding from analysis [19]. In addition, the study included a small number of solutions for each condition, which increased the likelihood that differences in dopaminergic activity associated with insight could be related to additional characteristics of problems or their solutions.

Another study tested the hypothesis that solving sudden insight problems is accompanied by neural reward. A high-frequency EEG was recorded while solving a series of anagrams. At each decision, the participants reported on the mechanisms for forming the response—as a sudden understanding or “analytically.” The participants then completed a questionnaire that assessed the overall dispositional sensitivity to reward. The frequency-time characteristics of the EEG were calculated separately for situations with solutions by the type of insight (I) and tests with analytical (A) solutions. Statistical parametric mapping (SPM) allowed the timing, frequency, and location of several effects to be established, where I > A. In contrast, no A > I effect was observed. The primary neural correlate of insight was a burst (I > A) in the vibrational activity of the EEG gamma range in the prefrontal cortex approximately 500 ms before the participants pressed the button indicating the solution to the problem [37].

A separate anterior prefrontal burst of EEG gamma activity was also recorded approximately 100 ms after the primary insight I-A effect, which was interpreted as a reward signal associated with insight. This interpretation was confirmed by reconstruction of the source showing that the signal was partially generated by the orbitofrontal cortex [37].

A limitation of the use of fMRI to study the mechanisms of neural reward associated with insight is its relatively low temporal resolution. To determine how much the brain response is integrated with or triggered by the insight, the technique must be able to show that the brain response occurs concurrently with or after the insight. A slow hemodynamic response, as measured by fMRI, is not an ideal tool for determining whether a brain response occurs immediately before, during, or after insight.

A line of studies has shown the presence of a neural reward signal associated with both insight and rewarding experience. This reward signal comes in part from the orbitofrontal cortex, a region associated with reward learning and subjective experiences of hedonic pleasure [17,21,24].

The results support fMRI data suggesting the presence of reward-related subcortical activity during insight solving and other studies in which limbic structures, in particular the amygdala, are involved in the processing of insights [15,17,37].

For investigation of the subcortical contribution to insight, ultrahigh-field fMRI was used. During the task, the subjects were offered triples of words and instructed to find a solution word associated with all three given words. They had to press the button as soon as they felt confident in their decision without further revision, which allowed them to capture the exact event—the Aha! moment. In addition to the detection of cortical involvement of the left anterior middle temporal gyrus, persistent changes in subcortical activity are associated with discerning problem solving in the bilateral thalamus, hippocampus, and dopaminergic midbrain, including the ventral tegmental region, nucleus accumbens, and caudate core. These results shed new light on the affective neural mechanisms underlying discerning problem solving [38].

In addition, there is a more general relationship between individual differences in the characteristics of the reward neural network and creative cognition [32,39].

The EEG, unlike fMRI, has a high temporal resolution. To this end, high temporal resolution of the EEG is used to isolate the neural reward signal from decision time and to show that its magnitude correlates with overall dispositional reward sensitivity. It has been found that people with a high sensitivity to reward perceive a sudden solution to the problem of awareness as a strong reward, while people with a low sensitivity to reward may perceive insight as sudden and attention grabbing but not hedonistic.

Thus, although the insight-related reward signal is generated close in time to the insight response, the reward signal does not appear to be the only necessary characteristic of insight. The short interval between insight and its associated reward signal suggests that this activity is not the product of a conscious appraisal of insight but rather is triggered by it. This finding is further supported by the fact that the amount of insight-related activity in the left temporal lobe was positively correlated with reward-related activity in the right orbitofrontal gyrus at the source level.

Seeking insightful solutions during the Remote Worker Assignment problem has been found to induce specific changes in the activity of the cerebral cortex. Given the strong affective components (Aha! moments) that manifest in subjectively experienced feelings of relief following the sudden appearance of a solution to a problem without any conscious warning, it has been suggested that the subcortical dopaminergic reward network is critically involved during insight.

Electrophysiological correlates of sensation of novelty in creative advertising were studied in 28 healthy subjects using ERPs. Participants viewed images that were difficult to interpret until a description was presented that contained either a creative description (CD) with an unexpected image description based on the original ad or a normal description (normal ND) that was a literal description of the image (and served as the initial condition). Participants rated the level of creativity in the description. The results showed that the amplitude of the N2 wave was higher for the CD than for the ND in the middle and right regions of the skull between 240 and 270 ms, which most likely reflects the detection of conflict. Moreover, the CDs showed a larger N400 value than the ND value over a time window of 380 to 500 ms, which is said to reflect semantic integration. This study investigated the electrophysiological correlates of novelty in advertising with environmentally sound incentives [39].

As expected, N2 (240–270 ms) and N400 (380–500 ms) were observed with higher amplitudes in the creative (CD) state than in the normal (ND) state. Moreover, creative stimuli were found to have a larger negative component (LNC; 500–700 ms) than normal stimuli. For the difference wave (CD-ND), the mean N2 amplitude caused a lot of negative activity in the scalp areas of the right and middle hemispheres; the mean amplitude of N400 was distributed with strong negative activity over the right hemisphere and the most posterior regions of the skull; the LNC amplitude is mainly distributed over the back of the scalp [40].

## 5. Conclusions

Thus, today we can talk about the following directions, methods of research on the phenomenon of insight, and the main research problems (see Figure 5). Among the areas of research are traditional ones—cognitive psychology, clinic, first of all, neurology, and psychiatry, as well as neurophysiology—in order to study the interaction of conscious and unconscious mechanisms of memory and learning. About the past decade, marketing research was added to the traditional directions—this is the peculiarity of the perception of advertising for brands and banking products, as well as research in the field of artificial intelligence and machine learning.

There are three groups of traditional neurophysiological research methods—EEGs, fMRI, and ERPs—with the basis of the P_300_ component. The authors have attributed to the advantages of EEGs the high temporal resolution, the registered EEG high-gamma activity in the right temporal lobe, as an indicator of the mechanism of binding of conscious information, and the time interval of the insight effect (1310–560 ms)—from the standpoint of making an analytical decision. The advantages of fMRI are high spatial resolution and registered increased blood flow in the right temporal lobe, hippocampus (as a reaction of pleasure), striatum, medial prefrontal cortex, and dopamine region—nuclei adjacent to the ventral region. The ERPs to a certain extent, combine the advantages of EEGs and fMRI: ERPs have high spatial and temporal resolution, allow a closer approach to the problem of insight scaling, the results allow us to evaluate internal and external insight, apart from the P_300_ component, and the components are important in the indication of insight over a period of time 500–700 ms.

The main research problems include, first of all, scaling (assessment) of advertising perception mechanisms, both direct (conventional) and indirect, where creative processes and cognitive processing are included. The decision to activate the mechanisms of cognitive processing (memory) represents the second group of problems. There are also questions about the indication of subjective sensations, emotional components associated with the enjoyment of insight, and the reorganization of the thought process based on the relationship between conscious and unconscious effects.

Thus, the main trends in the change in ERPs, depending on the variants of recognition of visual illusions and oddball images, are as follows. Complete recognition of illusions, which corresponded to insight in the model of our experiment, was accompanied by an increase in the amplitude of the P_300_ wave in the temporal-parietal regions on the right. With correct identification of one of the dual images, the activation of the N_450_ component was recorded in the frontal regions on both sides. Negative recognition was characterized by generalized symmetric activation of all ERP components, more pronounced in the posterior half of the brain.

A discussion of our results allows us to talk about two possible options for actualizing the mechanisms of long-term memory that ensure the formation of insight—the simultaneous perception of images as part of an illusion. The first of them is associated with the inhibition of the frontal cortex at the stage of synthesis of information flows, with the subsequent activation of the posterior parts of the brain and, most likely, directly related to the mechanisms of “extra-logical” thinking. The second variant is traditional and manifests itself in the activation of the anterior sections of the neocortex, with the subsequent excitation of all brain fields by the mechanisms of “exhaustive search.”

## Figures and Tables

**Figure 1 sensors-22-01323-f001:**
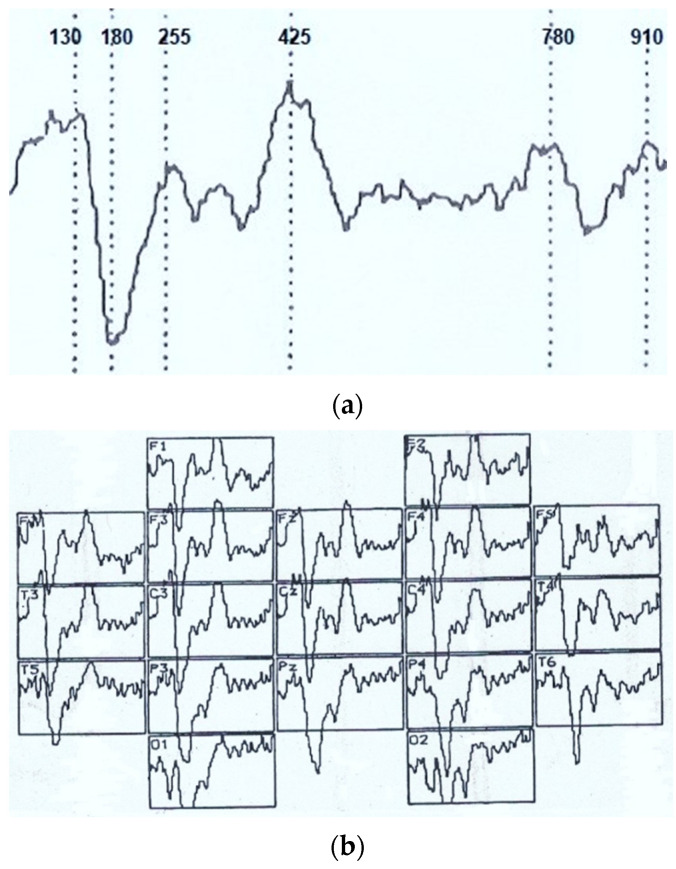
(**a**) The visual evoked potential (numbers—time, ms) and distribution of the VEP (**b**) according to the 10/20 system during the perception of an unrecognizable (oddball) visual image. Analysis time: 1000 ms.

**Figure 2 sensors-22-01323-f002:**
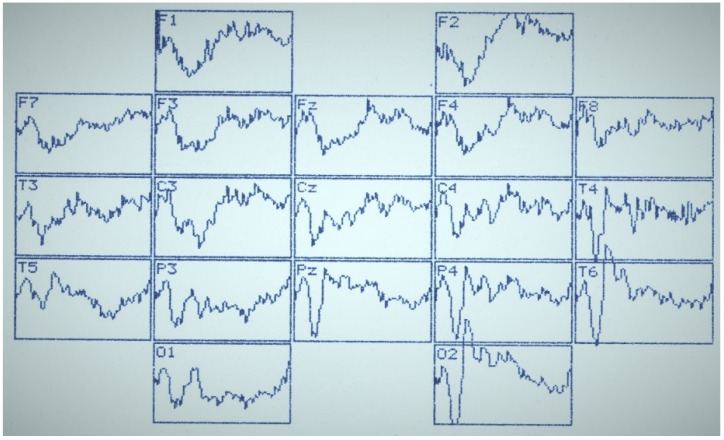
VEP distribution by the 10/20 system with correct recognition of images-illusions. Analysis epoch: 1000 ms.

**Figure 3 sensors-22-01323-f003:**
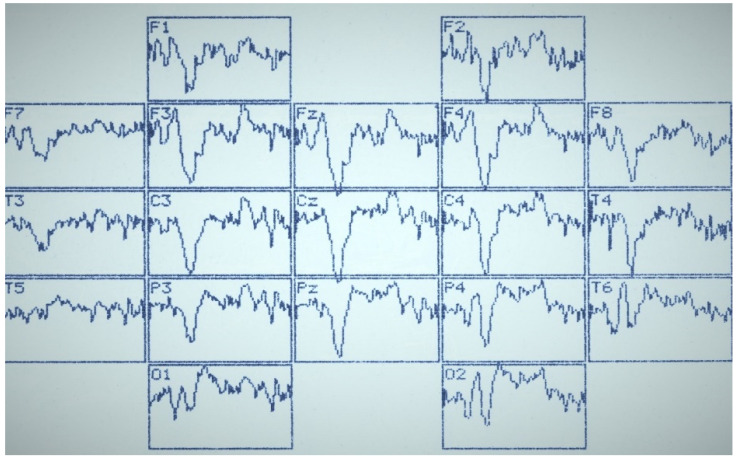
VEP distribution by the 10/20 system with correct recognition of one of the images-illusions. Analysis epoch: 1000 ms.

**Figure 4 sensors-22-01323-f004:**
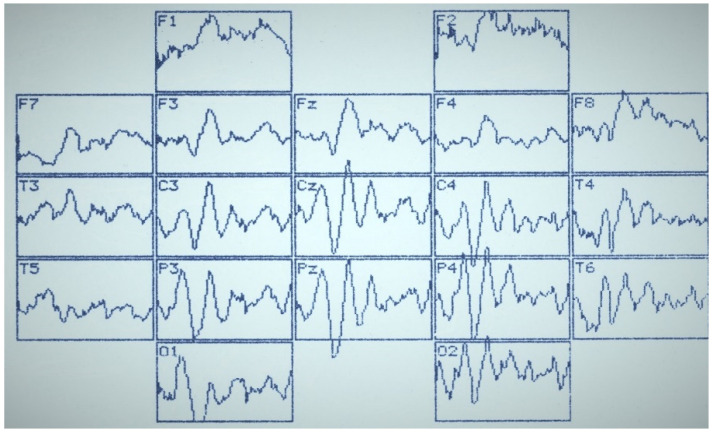
VEP distribution by the 10/20 system with unrecognition of images-illusions. Analysis epoch: 1000 ms.

**Figure 5 sensors-22-01323-f005:**
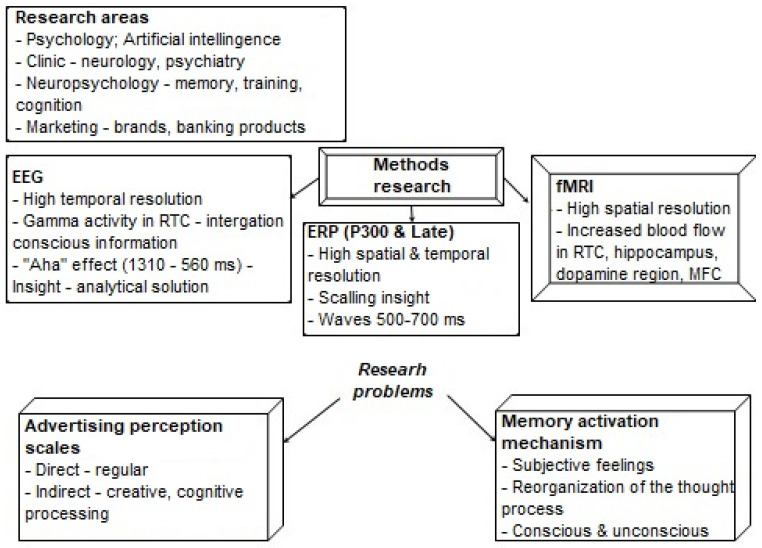
The general directions, methods, and problems of insight research.

**Table 1 sensors-22-01323-t001:** Quantitative characteristics of answer options in experiments with the presentation of dual visual images-illusions and correct identification of the oddball image.

Answer Options	Full Identification	Identification of One Image	Unrecognition	Oddball Image Identification
Number of responses	8	15	12	7

**Table 2 sensors-22-01323-t002:** Amplitude (A, uV) and time (T, ms) characteristics of the component N_150_ ERP.

Parameters	Sites	N_150_		
		Control	Recognition Illusions	Unrecognition Illusions
T	O2	161.3 ± 14.9	156.6 ± 22.9	
A		8.7 ± 3.4	7.4 ± 2.6	
T	O1	157.0 ± 15.6	146.5 ± 26.5	
A		10.4 ± 3.7	7.2 ± 3.8	n/a
T	Pz	144.0 ± 15.7	133.3 ± 18.6	
A		10.9 ± 3.5	12.6 ± 4.0	F < 4.0
T	C4	136.3 ± 12.7	133.3 ± 22.0	
A		14.2 ± 4.5	11.1 ± 1.7	
T	C3	132.6 ± 13.8	131.6 ± 15.0	
A		14.6 ± 4.3	9.4 ± 3.8 *	
T	F4	135.3 ± 14.9	135.8 ± 25.7	
A		11.7 ± 3.6	8.5 ± 3.0	
T	F3	131.0 ± 14.9	133.3 ± 17.5	
A		13.0 ± 3.2	8.6 ± 3.6 *	

Notes: T: time, ms; A: amplitude, uV; O2, O1…F3: sites on the 10/20 system; n/a: not applicable. The differences are not significant compared to the control group. The F-statistic value by compared to the control: * F > 4.0; in others, the differences are insignificant: F < 4.0.

**Table 3 sensors-22-01323-t003:** Amplitude (A, uV) and time (T, ms) characteristics of the component N_350_ ERP.

Parameters	Sites	N_350_		
		Control	Recognition Illusions	Unrecognition Illusions
T	O2	364.0 ± 59.3	291.6 ± 47.8 *	
A		2.8 ± 0.5	5.3 ± 2.4 *	
T	O1	371.6 ± 59.3	316.6 ± 46.6 *	
A		2.7 ± 0.8	4.1 ± 1.4 *	
T	Pz	374.0 ± 60.7	300.8 ± 57.3 *	
A		2.8 ± 0.8	4.6 ± 2.6	n/a
T	C4	371.3 ± 57.1	300.0 ± 68.8 *	
A		2.9 ± 1.2	4.5 ± 2.0	F < 4.0
T	C3	369.7 ± 56.0	335.0 ± 27.2	
A		2.9 ± 1.4	3.7 ± 1.1	
T	F4	367.6 ± 64.6	305.0 ± 77.3	
A		3.0 ± 1.1	4.6 ± 2.3	
T	F3	362.6 ± 65.5	302.5 ± 78.5	
A		3.2 ± 1.3	4.6 ± 2.1	

Notes: T: time, ms; A: amplitude, uV; O2, O1…F3: sites on the 10/20 system; n/a: not applicable. The differences are not significant compared to the control group. The F-statistic value by compared to the control: * F > 4.0; in others, the differences are insignificant: F < 4.0.

**Table 4 sensors-22-01323-t004:** Amplitude (A, uV) and time (T, ms) characteristics of the component N_450_ ERP.

Parameters	Sites	N_450_		
		Control	Recognition Illusions	Unrecognition Illusions
T	O2	461.6 ± 42.7	450.0 ± 43.3	461.6 ± 42.7
A		3.0 ± 1.3	4.1 ± 1.1	5.0 ± 1.3 *
T	O1	466.0 ± 37.9	460.0 ± 39.3	466.0 ± 37.9
A		2.9 ± 1.2	4.3 ± 1.6 *	4.9 ± 1.2 *
T	Pz	464.6 ± 40.3	460.8 ± 40.9	464.6 ± 40.3
A		3.5 ± 1.5	3.6 ± 2.3	5.5 ± 1.5 *
T	C4	459.7 ± 42.0	450.8 ± 44.5	459.7 ± 42.0
A		3.8 ± 1.8	4.3 ± 1.9	5.8 ± 1.8 *
T	C3	460.6 ± 40.8	459.1 ± 40.1	460.6 ± 40.8
A		4.0 ± 1.9	3.5 ± 2.2	5.0 ± 1.9 *
T	F4	456.2 ± 42.2	470.8 ± 59.0	456.2 ± 42.2
A		4.2 ± 1.5	2.8 ± 1.6	6.2 ± 2.5 **
T	F3	455.0 ± 44.3	460.0 ± 41.5	455.0 ± 44.3
A		4.5 ± 1.4	3.8 ± 2.1	6.5 ± 1.4 **

Notes: T: time, ms; A: amplitude, uV; O2, O1…F3: sites on the 10/20 system; n/a: not applicable. The differences are not significant compared to the control group. The F-statistic value by compared to the control: * F > 4.0; ** F > 10.0; in others, the differences are insignificant: F < 4.0.

**Table 5 sensors-22-01323-t005:** Amplitude (A, uV) and time (T, ms) characteristics of the component N_750_ ERP.

Parameters	Sites	N_750_		
		Control	Recognition Illusions	Unrecognition Illusions
T	O2	767.0 ± 71.3	724.1 ± 64.9	767.0 ± 71.3
A		2.5 ± 0.8	3.6 ± 1.3 *	2.3 ± 0.8
T	O1	774.6 ± 60.3	724.1 ± 77.0	774.6 ± 60.3
A		2.6 ± 1.0	2.9 ± 0.7	2.2 ± 1.0
T	Pz	761.8 ± 62.4	727.5 ± 65.4	761.8 ± 62.4
A		2.7 ± 1.0	3.3 ± 1.0	3.7 ± 1.0 *
T	C4	754.0 ± 70.1	725.0 ± 70.9	754.0 ± 70.1
A		2.5 ± 1.4	4.0 ± 1.6 *	3.5 ± 1.4 *
T	C3	766.3 ± 67.9	724.1 ± 76.2	766.3 ± 67.9
A		2.6 ± 1.3	3.5 ± 1.9	3.6 ± 1.3 *
T	F4	760.6 ± 64.3	717.5 ± 78.4	760.6 ± 64.3
A		2.1 ± 1.4	2.8 ± 0.5	4.1 ± 1.4 *
T	F3	762.3 ± 63.0	716.6 ± 73.8	762.3 ± 63.0
A		2.2 ± 1.0	3.0 ± 0.8	4.2 ± 1.2 *

Notes: T: time, ms; A: amplitude, uV; O2, O1…F3: sites on the 10/20 system; n/a: not applicable. The differences are not significant compared to the control group. The F-statistic value by compared to the control: * F > 4.0; in others, the differences are insignificant: F < 4.0.

**Table 6 sensors-22-01323-t006:** Amplitude (A, uV) and time (T, ms) characteristics of the component N_900_ ERP.

Parameters	Sites	N_900_		
		Control	Recognition Illusions	Unrecognition Illusions
T	O2	925.6 ± 38.0		925.6 ± 38.0
A		2.8 ± 1.2		2.4 ± 1.2
T	O1	922.6 ± 42.1		922.6 ± 42.1
A		2.5 ± 1.3	n/a	2.2 ± 1.0
T	Pz	921.0 ± 35.1		921.0 ± 35.1
A		2.8 ± 1.2		3.8 ± 1.2 *
T	C4	929.0 ± 27.3	F < 4.0	929.0 ± 27.3
A		2.3 ± 0.9		4.3 ± 0.9 *
T	C3	931.3 ± 31.3		931.3 ± 31.3
A		2.4 ± 0.9		4.4 ± 0.9 *
T	F4	929.6 ± 30.5		929.6 ± 30.5
A		2.1 ± 1.1		4.1 ± 1.1 *
T	F3	927.6 ± 30.1		927.6 ± 30.1
A		2.1 ± 1.6		5.1 ± 1.6 *

Notes: T: time, ms; A: amplitude, uV; O2, O1…F3: sites on the 10/20 system; n/a: not applicable. The differences are not significant compared to the control group. The F-statistic value by compared to the control: * F > 4.0; in others, the differences are insignificant: F < 4.0.

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
