# Peer review of "Long-Latency Event-Related Potentials (300–1000 ms) of the Visual Insight†"

_sensors, 2022, doi:10.3390/s22041323_

Round 1

Reviewer 1 Report

the paper could be interesting, but there are some points that must be addresses.

the abstract must be clear. The term Aha in the abstract is not understandable. line 19 disscus must be reviewed as discuss

what does the author mean with posterior part?  occipital?

and what the author mean with extra-logical thinking? please, explain better

 the first section of the introduction  (line 34-38) I think must be deleted, because it is repeated (in different form) in line 39.

the introduction could be more specific , introducing more example related the methodology (e.g., if there are some behavioral or psychophysiological study on the insight and the methodology used). the introduction is quite generic and don't allow to understand better the ratio to use the materials and methods proposed in the paper.

the research is conducted only in men. This is a great bias. So the author must also change the title of the paper. 

Caption of Table 2 isn't understandable, please, describe better

Image 5 is at low resolution. please. change  it with a high resolution

Author Response

Dear reviewer!

We are extremely grateful for your efforts in reviewing our work. We agree with most of your comments. Essentially, our answer is:

1. Changes have been made to the Annotation. The term Aha has been removed from the abstract. The term has been retained in the Discussion. In modern scientific literature, incl. rating journals, on the study of insight, the Aha is used as a term identical to insight. Although I agree that Aha is a popular science jargon.

2. Yes, the back of the brain is the occipital lobes. The term is used to avoid toftology. Replaced if possible. Additions have been made to non-logical thinking. Although, in essence, these are psychological mechanisms of insight.

3. Lines 34-38 removed.

4. In order to reach the goal setting, an analysis of the literature on insight over the past 20 years was carried out. For me, this problem is familiar even before. Little has changed in methodological solutions in neurophysiology over the past 20 years. These are EEG, fMRI and ERP. All the novelty of recent years is associated with posed problems and areas of application. Therefore, we have devoted the Introduction to these issues. And the methodology associated with the results is presented in the Discussion. This is also reflected in Fig. 5, original.

5. I don’t see the point in including gender features in the title. This is reflected in the methods. Moreover, we have data (not processed) for both healthy women and patients with brain pathology. For women, the trends in EP are similar, the difference is in the statistics of recognition of images.

6. Table notes have been expanded.

7. Resolution (dpi) has been increased for all pictures.

 With respect and best wishes.

Author

Reviewer 2 Report

Comments to the Author:

I have reviewed the paper entitled “Long-latency event-related potentials (300-1000 ms) of the visual insight”. In this study, author measured event-related potentials (ERP) under visual stimuli. The overall contents of this paper are well organized to give a clear overview of this work. It is an interesting research work about ERP with visual stimuli. I have some minor comments about this study are as follows:  

  1. The resolution of all figures1-4 is very low. Authors should write the x-axis and y-axis values/ captions.  
  2.  
  3. In method section, author should clearly explain about the pre-processing step of EEG signals , like how they remove artifact/ noisy 

    -What is sampling rate of EEG signals? 

    -Which filter author used for EEG signals?  

    -What is signal frequency range? 

    - What is reference channels name? 

    -What is the system/device name for EEG recording? 
  4.  
  5. -EEG cap with 32 channels or 64 channels or HD-EEG for recording EEG data? 
  1. 3. Author should add a flowchart regarding ERP analysis from step 1 to last step.
  2.  
  3. 4. The method and results sections are weak. Both should be revised clearly like how and why these results are novel with previous studies. Author can refer these latest papers based on ERP with visual stimuli, as following: 

-R. K. Chikara, O. Komarov L.W. Ko. “Neural signature of event related N200 and P300 modulation in parietal lobe during human response inhibition”, Int. J. Computational Biology and Drug Design, 2018 Vol. 11, Nos. 1/2, pp.171–182. 

  1. Authors should clearly explain the limitation of this study. 

Author Response

Dear reviewer!

We are extremely grateful for your efforts in reviewing our work. We agree with most of your comments. Essentially, our answer is:

  1. Resolution (dpi) has been increased for all pictures.
  1. Added to Methods - signal sampling rate - 500 Hz, amplifier bandwidth 0.5 - 30 Hz. These are the traditional parameters for recording long-latency EPs and EEGs. The purity of the signal was determined by automatic indicators. The registration of biosignals began with the registration of a traditional EEG. After that, the amplifier was switched to registration of super-averaged EPs. EPs are traditionally recorded unipolarly. With this registration, reference electrodes are placed on the earlobes. Accordingly, the dipole was the signal between each scalp and reference electrodes. We deliberately do not refer to the name of the manufacturer, as some rating journals consider this to be an advertisement. We worked on one of the international neurophysiological systems.
  2. At the preliminary stage, we considered options with 32 and 64 channels. 19 channels were chosen for this task (Fig. 2-4). Multichannel registration is more often used in EEG.
  3. Information (flowchart) on the analysis of EP is expanded in the text.
  4. We have added information from the featured article. Accordingly, article was included in the reference.
  5. There were no restrictions in this study. The work was carried out in accordance with the protocol of the ethical committee with informed voluntary consent. Limitations are usually detailed in the examination of patients.

With respect and best wishes.

Author

Round 2

Reviewer 1 Report

I accept the reviewed version